# Physical Activity, Dietary Patterns, and Glycemic Management in Active Individuals with Type 1 Diabetes: An Online Survey

**DOI:** 10.3390/ijerph18179332

**Published:** 2021-09-03

**Authors:** Sheri R. Colberg, Jihan Kannane, Norou Diawara

**Affiliations:** 1Department of Human Movement Sciences, Old Dominion University, Norfolk, VA 23508, USA; 2Department of Mathematics & Statistics, Old Dominion University, Norfolk, VA 23529, USA; jkann001@odu.edu (J.K.); ndiawara@odu.edu (N.D.)

**Keywords:** type 1 diabetes, A1C, physical activity, exercise, athletes, blood glucose, diet, CGM

## Abstract

Individuals with type 1 diabetes (T1D) are able to balance their blood glucose levels while engaging in a wide variety of physical activities and sports. However, insulin use forces them to contend with many daily training and performance challenges involved with fine-tuning medication dosing, physical activity levels, and dietary patterns to optimize their participation and performance. The aim of this study was to ascertain which variables related to the diabetes management of physically active individuals with T1D have the greatest impact on overall blood glucose levels (reported as A1C) in a real-world setting. A total of 220 individuals with T1D completed an online survey to self-report information about their glycemic management, physical activity patterns, carbohydrate and dietary intake, use of diabetes technologies, and other variables that impact diabetes management and health. In analyzing many variables affecting glycemic management, the primary significant finding was that A1C values in lower, recommended ranges (<7%) were significantly predicted by a very-low carbohydrate intake dietary pattern, whereas the use of continuous glucose monitoring (CGM) devices had the greatest predictive ability when A1C was above recommended (≥7%). Various aspects of physical activity participation (including type, weekly time, frequency, and intensity) were not significantly associated with A1C for participants in this survey. In conclusion, when individuals with T1D are already physically active, dietary changes and more frequent monitoring of glucose may be most capable of further enhancing glycemic management.

## 1. Introduction

In 2021, a full century has passed since the 1921 discovery of insulin [1], a hormone that must be replaced in individuals with type 1 diabetes (T1D), all of whom have lost the ability to produce it as the result of primarily autoimmune destruction of the pancreatic β-cells [2]. Since its discovery, replacement insulin has evolved greatly with numerous types and delivery methods now possible, along with use of better glycemic management and tracking tools that can assist individuals in preventing acute and chronic diabetes-related health complications. In fact, most people with T1D can expect to experience near normal longevity with a high quality of life, particularly if glycemic management and cardiovascular health are maintained [3].

When undertaken by individuals of all ages with T1D, physical activity is associated with many well-established health benefits, including improved cardiovascular fitness, lower cardiovascular risk, better quality overall health, and enhanced psychological well-being [4,5]. One of the major factors linked with their long-term survival is the absence of features of the metabolic syndrome and, more specifically, the presence of insulin sensitivity [6]. Physical activity of all types has been associated with greater insulin sensitivity [7,8,9]. In adults with T1D, being regularly active improves cardiometabolic risk profile [10] and is associated with increased longevity [6,11]. Individuals with T1D of all ages are capable of engaging in a wide variety of physical activities and sports, ranging from recreational to Olympic-level (12), and many choose to be physically active to achieve unique goals related to athletics and/or health. However, these individuals must contend with the continuous challenges associated with being physically active with T1D, including monitoring glucose levels, managing dietary choices and intake, adjusting insulin doses, and adapting daily regimens to account for other factors that impact glycemia [12,13,14,15]. Physical activity acutely can lead to hypoglycemia and hyperglycemia [15,16,17,18,19,20], either of which may become a medical emergency if not adequately managed.

Numerous physical activity training patterns and regimens are possible with T1D, and each individual must choose to follow the one that works uniquely best, although that may vary with the type, intensity, frequency, and timing of activities, among other variables [18,19,21,22]. High-intensity training as well as competition can substantially increase glucose output from the liver, potentially leading to hyperglycemia both before and during activity [19]. Resistance exercise is associated with less of a decline in blood glucose than aerobic [18,23] and can provide a protective effect against glycemic declines if performed prior to aerobic exercise [24]. Even the timing of exercise can impact outcomes, with exercise before breakfast resulting in less hypoglycemia than the same bout of aerobic or resistance activity undertaken later in the day [18,23,25]. An appropriate dose of rapid-acting insulin can be used to treat hyperglycemia after morning exercise of any type without inducing hypoglycemia post-exercise [26]. In addition, exercise glycemic management strategies often vary within sporting events [27,28] and afterward [20,28]. 

In addition, nutrition and dietary patterns are one of the more controversial topics related to athletic performance in all individuals, as well as to glycemic management in T1D and overall health [12,13,29,30,31]. Whether individuals are participating in sports and activities recreationally or aiming for competitive levels of athletic achievement, their performance can be positively or negatively impacted by a number of nutritional factors, such as intake and timing of macronutrients, availability of micronutrients, hydration status and electrolyte balance, and exercise training practices [12,13,14]. In particular, carbohydrate consumption to fuel the exercise bout and/or for hypoglycemia prevention is an important cornerstone to maintain performance and avoid hypoglycemia [31,32].

Use of some of the insulin delivery systems, glucose monitoring devices, algorithms, other glucose-focused technology and tools may also improve how well activity can be managed [26,33,34,35]. Recent technological advances, such as insulin pumps and continuous glucose monitoring (CGM) devices, have greatly advanced the ability of individuals to manage glucose levels around physical activity by allowing for almost real-time changes in insulin delivery and feedback on glycemic responses [36,37]. When using continuous subcutaneous insulin infusion (i.e., an insulin pump), active individuals can reduce or suspend basal insulin infusion at the start of exercise [38], or even starting 30–60 min before exercise [39], in order to mitigate declines in blood glucose. Likewise, CGM devices have been shown to improve glycemic management [40,41,42], even in individuals with T1D with lower A1C (a measure of overall blood glucose over the last 2–3 months) values already [43]. However, CGM measure glucose in interstitial spaces, and a time lag exists between blood glucose (measured via finger stick) and CGM glucose levels (measured via CGM) [36,44,45], making it unclear whether use of such devices can benefit glycemic management with physical activity. Finally, integrated insulin pump and CGM systems have shown promise with regard to ameliorating glycemic management in individuals with T1D [35,46,47,48], but their successful use around exercise remains more limited [49,50,51,52,53,54]. 

Thus, the purpose of this study was to ascertain which variables related to the diabetes management of physically active individuals with T1D have the greatest impact on overall blood glucose levels (via A1C) in a cohort of active adults and adolescents with T1D in a real-world setting. Given the complexity of managing blood glucose levels when exogenous insulin must be precisely balanced with food intake for any physical activity, we hypothesized that both physical activity (total weekly time, frequency, intensity, and/or type) and dietary patterns (particularly carbohydrate intake) would potentially impact overall blood glucose management in these active individuals, along with the use of the latest diabetes technologies (e.g., insulin pumps and CGM devices).

## 2. Materials and Methods

### 2.1. Subject Recruitment

An online survey conducted in English was advertised in 2018 by investigators on diabetes-focused social media platforms and distributed to various professional contacts via email. Participation was completely voluntary with no incentives offered, and the survey was open to all physically active individuals with diabetes of any age during a month-long period. The survey itself was completed through a separate online platform and contained no questions that could be used to identify personal data or characteristics by the investigators. Data collection methods were considered exempt from requiring participant consent by our university due to the online anonymous and voluntary nature in which all survey responses were obtained and recorded.

A total of 220 participants (109 male, 111 female, age range of 13 to 84 years) who had been diagnosed with T1D for varying lengths of time were included in the study. Their distributions by age and years with T1D are shown in Figure 1.

### 2.2. Online Survey and Data Collection

The online survey included a broad array of questions that participants could choose to complete with none being mandatory. Self-reported data about each participant included the following variables: age, sex, diabetes type, latest A1C value, usual insulin regimen (including insulin pumps), use of other medications, glucose self-monitoring practices (i.e., frequency and use of CGM devices), typical dietary patterns and estimated carbohydrate intake, physical activity patterns, target blood glucose ranges for exercise, regimen changes for physical activity, and typical treatments for exercise-related hypoglycemia or hyperglycemia. Any A1C values that were reported in mmol/mol (all coming from respondents outside the United States) were converted to equivalent % values before analysis, and only self-reported insulin users were included in the analyses.

#### 2.2.1. Physical Activity Participation and Categorization

Physical activity participation was assessed with questions about typical frequency, intensity, time, and type. Their usual intensity was self-categorized as light, moderate, vigorous (hard), very hard, or maximal using drop-down selections found in the survey. Total physical activity time per week was calculated as a product of self-reported days of activity per week and the typical amount of time spent exercising per day regardless of the activities undertaken. Additional open-ended responses related to participants’ individualized diabetes regimen changes were collected for over 165 different sports and activities, which were largely used for other purposes [55]. Responses to these physical activity and other related, open-ended questions were not directly analyzed and only included in terms of whether participants reported engaging in various activities. 

Participants’ self-reported activities were placed into one or more of five categories: fitness, endurance, endurance-power, power, and outdoor. The designation of each sport was determined by the investigators and primarily based on the energy systems engaged during the activity itself (aerobic vs. anaerobic ones) [56], although some overlap among categories exists for certain sports and activities. Once participants answered “yes” for a category, numerous examples of activities and sports in each category were provided in the survey as drop-down selections to steer them to select representative ones. Some examples of selections in each category included, but were not limited to, the following:Fitness activities: fitness walking, aerobic conditioning machines, resistance training, aerobics classes, Pilates, kettle ball training, dancing, agility training, balance training, stretching, yoga, indoor climbing, martial arts, tai chi, physical activity classes;Endurance activities/sports: running and jogging, swimming, cycling, marathons, biathlons, triathlons, cross-country running or skiing, ultra endurance training;Endurance-power sports: basketball, soccer, golf, tennis, hockey, football, tennis, indoor racquet sports, intermediate-distance track events, CrossFit, high-intensity interval training;Power sports: baseball, bodybuilding, Olympic weight lifting or power lifting, sprinting, field events (shot put, pole vault, high jump, etc.), volleyball or beach volleyball;Outdoor activities/sports: kayaking, downhill skiing, curling, waterskiing or wakeboarding, kiteboarding, hiking and backpacking, horseback riding, rock or ice climbing, adventure racing, trail running, hunting, fishing, gardening, etc.

#### 2.2.2. Dietary Patterns and Carbohydrate Intake

The usual dietary patterns of participants were assessed with specific questions about whether they ingested carbohydrate for physical activity, their preferred sport-specific carbohydrate choices, and their usual dietary treatments for hypoglycemia, along with more open-ended questions about their typical dietary patterns. Some responded with definitive dietary patterns from which carbohydrate intake could be easily estimated, such as “keto diet” [57] or “Dr. Bernstein diet” [58,59], whereas others gave actual daily carbohydrate estimates or stated that they were vegan or vegetarian, ate a meat-based diet, consumed a plant-based whole foods diet, or avoided/limited their intake of starches or other food categories. These carbohydrate intake/dietary pattern data have been reported for a larger cohort of individuals with T1D or type 2 diabetes previously [12,13]. All of their responses to nutrition-related or dietary questions were considered together by the investigators, along with typical calorie requirements for active adults and adolescents [60], when estimating participants’ generalized daily carbohydrate intake and placing them into one of four categories for analyses: Normal (unrestricted): >200 g/day;Moderate: 100–200 g/day;Low-carbohydrate: 40–99 g/day;Very low-carbohydrate: <40 g/day.

### 2.3. Statistical Analyses

For this study, descriptive variables are presented as mean, standard error of the mean (SE), median, minimum, and maximum. A generalized linear model (GLM) approach was used to measure and quantify association between A1C and predictor variables. Using GLM, the equation for these associations was formulated as:$$y_{i}=\beta_{0}+\beta_{1}x_{1i}+\beta_{2}x_{2i}+\ldots +\beta_{p}x_{pi}+e_{i},\;$$
where $y_{i}$ represents the response of the *i*th participant’s A1C, for i=1,…,n, with $x_{1},x_{2},\ldots ,x_{p}$ representing other predictors like biological sex, usual carbohydrate intake, use of CGM devices, and other collected variables. Predictor variables were either discrete or continuous. In the model equation, the term $\beta_{0}$ served as the model intercept and $\beta_{i}$ referred to the slope associated with the $i^{th}$ predictor variable, with the errors $e_{i}\;\mathrm{independent}\;\mathrm{and}\;\mathrm{identically}\;\mathrm{distributed}\;\sim N\left(0,\;\sigma^{2}\right)$ and $\sigma^{2}$ with the model variance. In order to minimize variance and satisfy model assumptions, a transformation of the A1C to a natural log scale was applied. 

Due to a gap in self-reported A1C values, the natural log values (log A1C) were found to be closer to the normal distribution than the A1C itself. Consequently, log A1C values were used for further analyses in the GLM (see Appendix A for a detailed justification of the transformation to natural log and results of statistical tests). Significance for all such analyses was set as *p* < 0.05.

## 3. Results

### 3.1. Participant Characteristics and Survey Responses 

The demographic factors of participants included their A1C, age, and years living with diabetes, as shown in Table 1, along with responses to other quantifiable and categorical questions from the online survey. The majority of the 220 respondents were from the United States (68%), with others from Europe (13%), Canada (7%), Australia (6%), Eastern Europe (3%), and the rest (3%) from Mexico, South Africa, Iran, India, and the Philippines. Data from another 30 participants with T1D were excluded due to incomplete or missing responses related to A1C and other relevant variables.

As a whole, the participants’ latest A1C mean and median values (Table 1) were well within commonly recommended ranges of less than 7% [61]. Almost 70% self-reported having an A1C within this recommended range, although values ranged from 4.2% to 10.5%. About 25 individuals reported using a second diabetes medication besides insulin, with the majority of them using either metformin or a sodium-glucose transport protein 2 (SGLT2) inhibitor. As none of these medications impacts exercise-associated blood glucose levels, they were not included in any further analyses.

### 3.2. AIC and Its Predictors

#### 3.2.1. A1C Prediction with Physical Activity Variables 

The total weekly time spent being physically active was estimated based on participant responses to both frequency (number of days per week) and usual time spent exercising on active days. The total hours per week were calculated as a product of the two, and the distribution of participant time is shown in Figure 2. The nature of the survey did not allow for any differentiation among time spent doing different types of activities.

When total physical activity time per week was further categorized into whether participants’ met the recommended minimum (at least 150 min, or 2.5 h, of aerobic activity) or engaging in less than 150 min [62,63], total time was not significantly predictive of log A1C values regardless of whether or not participants met weekly physical activity recommendations (Figure 3). Total time, however, included all types of activities in this survey.

The number of days of activity per week ranged from 2 to 7 (Figure 4), demonstrating that all participants were physically active. However, frequency of physical activity was not a significant predictor of A1C. 

The usual intensity of physical activity engaged in by participants ranged from light to maximal, depending on the sport or activity (Figure 5). However, intensity also failed to predict differences in A1C values.

#### 3.2.2. A1C Prediction with Categorical Responses to Selected Survey Responses 

Survey responses related to participation in each category of physical activity or sports and carbohydrate ingestion for activity are shown in Figure 6 and Figure 7. No significant associations were found between these categorical responses and log A1C for any of these.

#### 3.2.3. A1C Prediction Based on CMG Use and Dietary Patterns 

CGM device use and whether participants experience activity-related low and high blood glucose values are shown in Figure 8. No significant associations were found between these yes/no categorical responses and log A1C for these variables.

With all variables considered within our model, the only significant predictors of participants’ log A1C values ended up being their use of CGM devices (*p* = 0.02) and their typical carbohydrate intake (*p* < 0.0001). These associations remained strong when analyzing either A1C or transformed natural log A1C (analyses shown in Appendix A). However, the variance was significantly reduced when the prediction model used log A1C given the gap evident in the distribution of participants’ A1C values (see Figure A1). The overall associations between A1C and usual carbohydrate intake categories are shown in Figure 9, and the relative percentages of participants falling into each carbohydrate intake category are shown in Figure 10.

#### 3.2.4. A1C Prediction Based on Attainment of Recommended Ranges 

An even more precise prediction emerged when participants were separated into one of two groups based whether their A1C values fell into the recommended range (<7%) or above (≥7%). A very-low carbohydrate intake was significantly associated with the lowest log A1C values when in recommended ranges (*p* < 0.0001), but CGM use was not predictive in that case (*p* = 0.90). When log A1C was in ranges above recommended, the most significant predictor was CGM wear (*p* < 0.01), with users of the devices having significantly lower values even though they failed to meet A1C recommendations, although carbohydrate intake failed to be predictive when A1C was higher (*p* = 0.16).

## 4. Discussion

As the aim of this online survey study was to ascertain which variables related to the diabetes management have the greatest impact on overall blood glucose levels, the outcomes were focused around achievement of A1C values in a recommended range. The primary findings were that in this cohort of free-living, physically active individuals with T1D of various ages, lower A1C values (within the recommended range of <7%) were best predicted by following a very-low carbohydrate dietary pattern, whereas using a CGM device was associated with better A1C values when A1C was higher than recommended. Contrary to our expectations, participants’ self-reported physical activity levels were not predictive of A1C values, even when they engaged in recommended amounts of total weekly activity of any type, and consideration of frequency, intensity, type, or total time did not increase the predictive value. However, most participants were already very active when compared to the population as a whole, which likely impacted these findings.

Reliance on physical activity participation to better manage overall blood glucose in individuals with T1D has shown mixed outcomes, although recent results are more promising (21). Participants in our online survey reported a fairly wide range of A1C values, demonstrating that being physically active alone does not guarantee optimal glycemic management, although the majority of values (70%) fell in the recommended range of <7% and would be considered well-managed. These results concur overall with many other studies showing that unless other glycemic variables are effectively balanced at the same time—such as food intake, insulin doses, and physical or mental stress—individuals with T1D do not necessarily experience improvements in overall glucose values when regularly active, with some studies demonstrating benefits [64,65,66] and others finding no improvement in A1C following aerobic or resistance training [67,68]. Our participants were engaging in myriad activities, though, making interpretation more difficult compared to those studies and others in which activities were more controlled and uniform. Moreover, our survey respondents engaged in physical activity 2 to 7 days per week, with over 93% of them reportedly engaging in more than the minimal recommended time. Some were training up to 42 h of weekly as competitive athletes and only five participants were active less than 100 min per week. This level of participation is far more than in the population overall [69,70] and for most with diabetes [71]. While our survey was not capable of discerning time spent in aerobic (as recommended 150 to 300 or more minutes a week) versus other types of activities, others have shown that total exercise volume and time spent being physically active doing any type of activity may matter more to cardiovascular and metabolic health than participation in specific bouts of moderate-to-vigorous aerobic activities by themselves [72,73,74,75]. Engaging in muscle-strengthening activity ≥2 times/week may provide additional benefits among insufficiently active adults [76]. With these observations in mind, we felt comfortable categorizing our participants as meeting or failing to meet the recommended total activity time with all types of activities considered together, not just aerobic ones.

Being physically active with T1D increases an individual’s risk of activity-related hypoglycemia [77,78,79] and hyperglycemia [80,81], and fear of activity-related hypoglycemia has often been a deterrent of regular participation for insulin users of all ages [82,83]. Conversely, since all of our participants were engaging in regular physical activity, they likely had already adapted their diabetes management strategies to better manage their glycemic variations while minimizing any fear of hypoglycemia associated with being active; in fact, out of 220 participants, only 10 reported A1C values of 8% or higher and only two of those were above 9%. Their regular participation may also at least partly explain why their total activity was not predictive of overall glycemic management since the vast majority were already exceeding recommended levels of activity and had A1C values that were well-managed compared to the majority of individuals with T1D as a whole [41,42]. Thus, it is likely that the glycemic impact of being active was already reflected in their having better A1C values than most individuals with T1D.

Another challenge associated with attempting to achieve better A1C values with physical activity is the unpredictability of glucose responses even to similar bouts of exercise. Active individuals completing our online survey frequently expressed frustrations with maintaining glycemic balance while doing a variety of physical activities under free-living conditions [55]. A recent study conducted on 12 adults with T1D reported that three identical cycling sessions completed on separate days under controlled conditions resulted in varying values for glucose measured either with a finger-stick (capillary blood) blood glucose monitor or a CGM device, even though glucose declined in all three trials [78]; these results indicated low reproducibility at the participant level and remained unchanged after adjustment for baseline glucose values. Likewise, in adolescents with T1D, while greater intrasubject reliability and repeatability of blood glucose responses to prolonged exercise was shown to be possible, this result occurred only when pre-exercise meal, exercise, and insulin regimens were kept constant [84], which is not always feasible in real life. However, recent technological advances, improvements in insulin regimens, newer insulins, and a better understanding of the physiology of various types of exercise may help limit such unpredictability for similar activities and, at the same time, lessen the fear of hypoglycemia by facilitating hypoglycemia prevention [82]. With proper management around activities, athletes with T1D at all levels have been shown to be capable of undertaking and performing well even in long endurance training, high-intensity sports, and other types of events [27,29,85,86].

With regard to dietary patterns, in the current study a very-low carbohydrate intake was surprisingly most predictive of achieving recommended glycemic levels overall (i.e., an A1C < 7%), regardless of differing levels and types of physical activity participation. Many endurance athletes with and without T1D have claimed to perform well with a lower, or at least moderate, intake of this macronutrient [57,87] and to maintain a better glycemic balance [31], although the consensus remains that carbohydrates are necessary to perform well at higher intensities and durations of activity [12,88,89]. However, the active individuals in our study who stated that they ingest carbohydrates during physical activity had similar A1C values to those who claimed to refrain from carbohydrate supplementation. In fact, supplementing with carbohydrates has been shown to potentially be superior to bolus insulin reduction for prevention of hypoglycemia during physical activity, as was demonstrated in a group of adults with T1D engaging in moderate-intensity cycling for 45 min in one study [90]. In our survey, however, both strategies (i.e., carbohydrate ingestion and insulin reduction) were used by most participants to prevent hypoglycemia with activity; in many cases, active individuals with T1D must employ a combination of both in order to maintain glycemic balance during and after training or events [12,15,29]. 

Nowadays, daily carbohydrate intake alone is usually not predictive of A1C values for most with T1D, and consuming carbohydrates can be feasible, which may be reflective of individuals’ use of faster-acting insulin analogues for meal boluses. In fact, a single mealtime bolus of insulin has been shown to cover a range of carbohydrate intake without deterioration in postprandial glycemia [91]. Even dietary fat, protein, and the glycemic index of ingested carbohydrates are associated with insulin dosing needs and impact postprandial glucose excursions [92,93], making glycemic predictions and insulin dosing based on grams of carbohydrate intake alone inadequate. Carbohydrate counting is fraught with complications given the complexities in digestion and absorption rates of that macronutrient and challenges related to proper estimation of the amount ingested by individuals [94,95,96]. With regard to our survey participants, most of whom already had optimal blood glucose management, it may be that they simply were able to tighten it slightly further by restricting their carbohydrate intake. Avoiding greater fluctuations in blood glucose after meals and during activity can improve overall glycemia [97]. In our survey, for individuals with higher-than-recommended A1C values, carbohydrate restriction was not predictive of better glycemic management, suggesting that other variables are impacting glycemia more in their case. 

Although trials are undergoing, to date low- and very low-carbohydrate diets have not been extensively studied in the management of T1D [13], with available studies examining glycemic outcomes from such diets being largely cross-sectional and lacking validated dietary data or control subjects [32,98]. Many of the participants in such studies can be described as highly motivated individuals who follow intensive insulin management practices, including frequent blood glucose monitoring and additional insulin corrections to meet tight glycemic targets. While athletes may still perform adequately when following such restricted diets [32,99,100], some potential negative health consequences of ketogenic and other low-carbohydrate diets have been noted [101,102], and longer term studies are needed to determine how feasible these dietary patterns are for most individuals with T1D [103]. Thus, much work remains to be done to fully determine the extent of the impact of dietary carbohydrate restriction on glycemic outcomes and optimal intake levels, particularly in physically active individuals with T1D.

Finally, the use of the latest diabetes technological advances, such as insulin pumps and CGM devices, has greatly advanced the ability to manage glucose levels around physical activity [36,37]. While 60% of our participants used an insulin pump, an even larger percentage (77%) used a CGM. Having access to either one or both devices potentially can allow users to make more informed choices to manage glycemia around exercise [104]. For our survey participants, using a CGM device was predictive of lower A1C values (specifically when above recommended levels) although insulin pump use was not predictive. This is unsurprising given that other studies have shown that CGM can be beneficial for all individuals with T1D [40,41,42], even for those who have already achieved recommended A1C at <7.0% [43]. Despite the demonstrated time lag between blood glucose (measured via finger stick) and interstitial glucose levels (measured via CGM) [36,44,45], having closer to real-time feedback on the impact of any activity likely makes glycemic management easier, especially when activities can vary so widely in their effects. For instance, a recent systematic review and meta-analysis that included 12 studies using CGM devices to examine the delayed impact of engaging in various physical activities reported that intermittent exercise (i.e., most endurance-power or power sports) actually increases the time spent in hypoglycemia and lowers mean glycemic values via CGM, with no differences in time spent in hyperglycemia or the number of hypoglycemic events [105]. Hypoglycemia risk was also lower for activities performed in the morning rather than in the afternoon, even with a 50% rapid-acting insulin reduction prior to later-day exercise. While our participants did not indicate their usual time of day for activities, CGM use has the potential to provide feedback that allows users to take corrective actions to manage glycemia in a timelier manner. 

Although not a survey question, some of our participants noted employing various exercise strategies with use of hybrid closed-loop systems (i.e., Medtronic 670G), which involve integration of an insulin pump, CGM, and algorithm control system to manage insulin delivery in response to real-time glucose levels with minimal user input. Although some input is usually still required (such as announcement of meals or exercise), hybrid systems have recently been found to improve time-in-range (typically defined as 70–180 mg/dL, or 3.9–10.0 mmol/L) around physical activity [106]. Users of such systems with a lower intake of daily carbohydrates have also experienced better glycemic management [107], likely due to the ability of such systems to make adjustments in response to the slower glucose fluctuations resulting from dietary protein and fat [97,108].

The limitations of this survey research localize mainly around our inability to collect more quantifiable and directly verifiable data, since all of it was self-reported and many of the survey questions were more open-ended. This is particularly an issue for dietary considerations including estimating carbohydrate intake, total calories, macronutrient distribution, and micronutrient adequacy, among other considerations. The authors used their best judgment when placing the participants into dietary categories for carbohydrate intake based on the data collected. However, it is possible that their interpretation of some responses was flawed or that participants failed to report or recognize all the carbohydrate sources in their diets, including those in high-fat, low-carbohydrate foods (e.g., olives, avocados, and nuts); in foods, drinks, or sports supplements taken during activities; and in rapid hypoglycemia treatments. A dietary recall questionnaire would have enhanced the reliability of these data around dietary patterns, total calorie intake, and macronutrient distribution. Likewise, although participants responded to questions around insulin use, types, and delivery methods, our interpretations are limited. More information related to actual dosing, timing, and other insulin-related data, particularly around physical activity and glycemic management would have provided more definitive results. Finally, relying on self-reported data in any research study has its limitations and can be problematic [109,110]; this is particularly true when it comes to data related to physical activity. Our survey participants reported engaging in a wide array of physical activities, many of which have varied glucose responses even within a specific category, especially “outdoor activities and sports”. Our data collection and interpretation would have been enhanced by use of a more standardized physical activity questionnaire, quantifiable data that could be converted into objective total exercise volume measures (such as MET-min/week) and, of course, controlled laboratory conditions.

Much remains to be studied related to physical activity in individuals with T1D, especially given the large number of variables that must be simultaneously balanced to maintain normal or near normal glycemic levels. Future research likely should include the potential implications of carbohydrate-restriction and other dietary patterns on physical activity performance and glycemic balance in this population. Another area to pursue is the glycemic benefits of using the latest technologies related to insulin delivery, glucose monitoring, and physical activity trackers and other devices. Such technologies can provide immediate feedback to users and allow them to make optimal and real-time diabetes regimen adjustments before, during, and after physical activity. 

## 5. Conclusions

In conclusion, when individuals with type 1 diabetes of any age are already physically active and their blood glucose is well-managed, a greater focus on lowering carbohydrate intake may improve glycemic management. In addition, active individuals may benefit from using continuous glucose monitoring to lower overall glycemia, especially when their A1C values are higher than recommended. Nevertheless, all individuals can benefit from being physically active on a regular basis, especially when the myriad variables affecting glucose responses can be adequately managed to prevent hypoglycemia or hyperglycemia.

## Figures and Tables

**Figure 1 ijerph-18-09332-f001:**
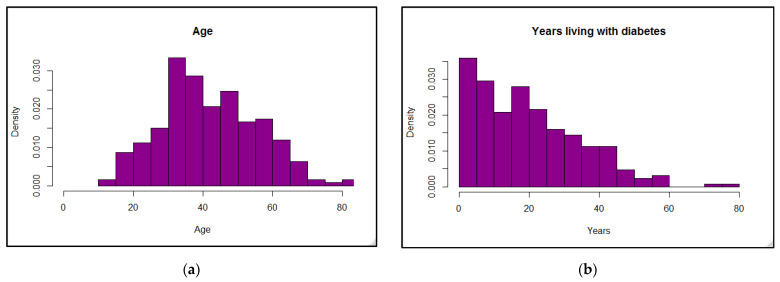
Distribution of participants by age (**a**) and years living with type 1 diabetes (**b**).

**Figure 2 ijerph-18-09332-f002:**
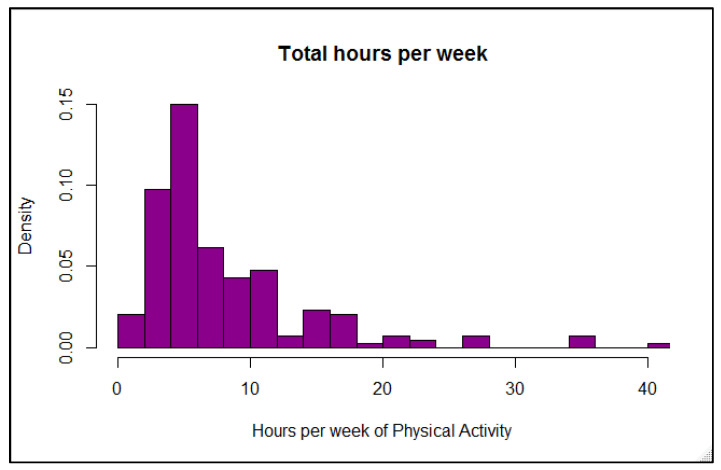
Distribution of participants by total hours per week spent doing all physical activities.

**Figure 3 ijerph-18-09332-f003:**
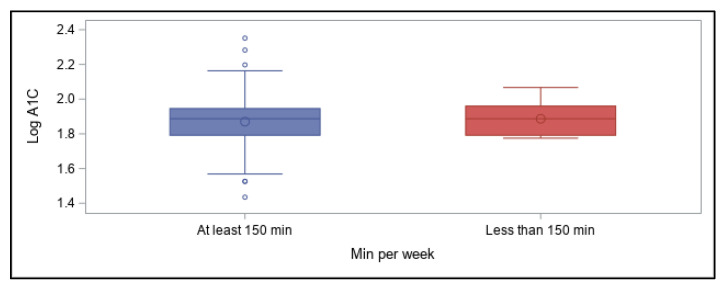
Total physical activity time by recommended amount and association with log A1C values.

**Figure 4 ijerph-18-09332-f004:**
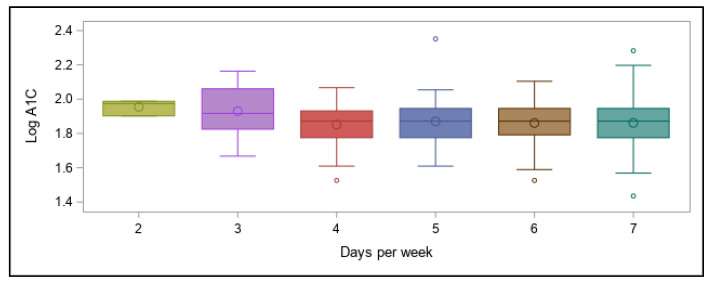
Number of days of physical activity per week and association with log A1C values.

**Figure 5 ijerph-18-09332-f005:**
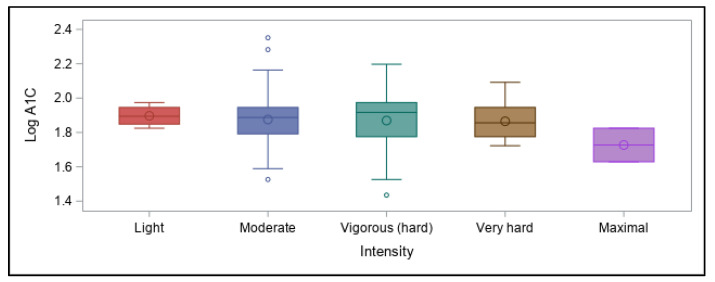
Intensity of physical activity and association with log A1C values.

**Figure 6 ijerph-18-09332-f006:**
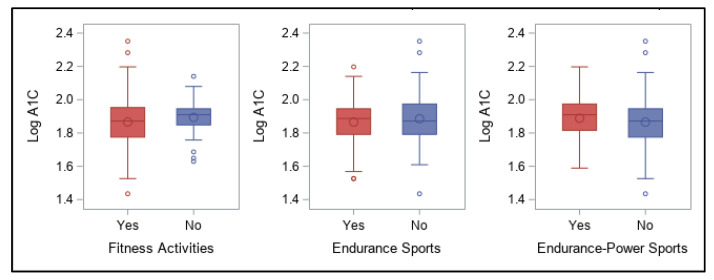
Participation in fitness activities, endurance sports, and endurance-power sports and association with log A1C values.

**Figure 7 ijerph-18-09332-f007:**
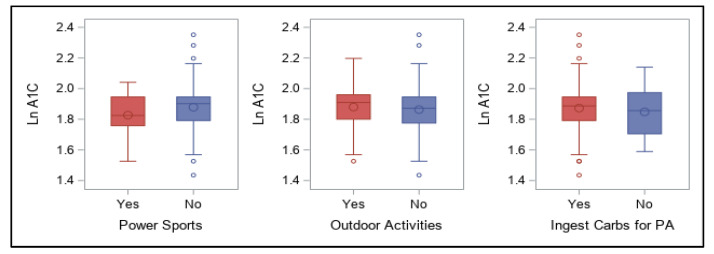
Participation in power sports and outdoor activities and ingestion of carbohydrates for physical activity and association with log A1C values.

**Figure 8 ijerph-18-09332-f008:**
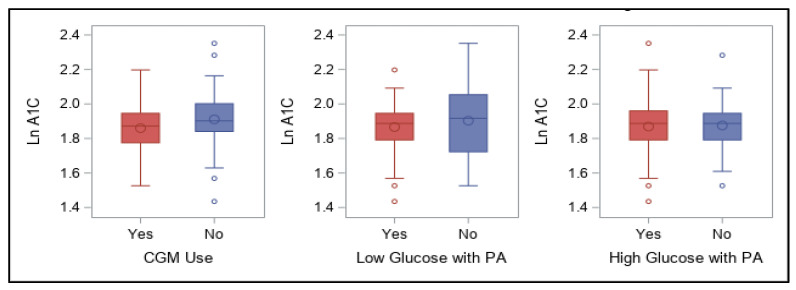
Use of CGM and physical activity-related low and high glucose and association with log A1C values.

**Figure 9 ijerph-18-09332-f009:**
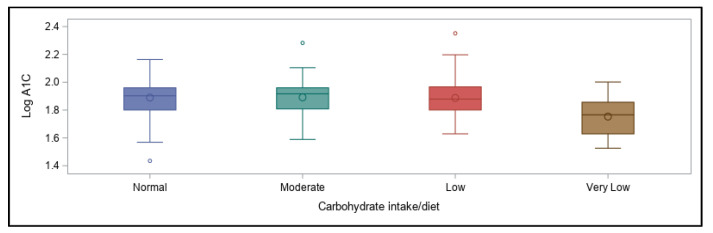
Usual daily carbohydrate intake and association with log A1C values.

**Figure 10 ijerph-18-09332-f010:**
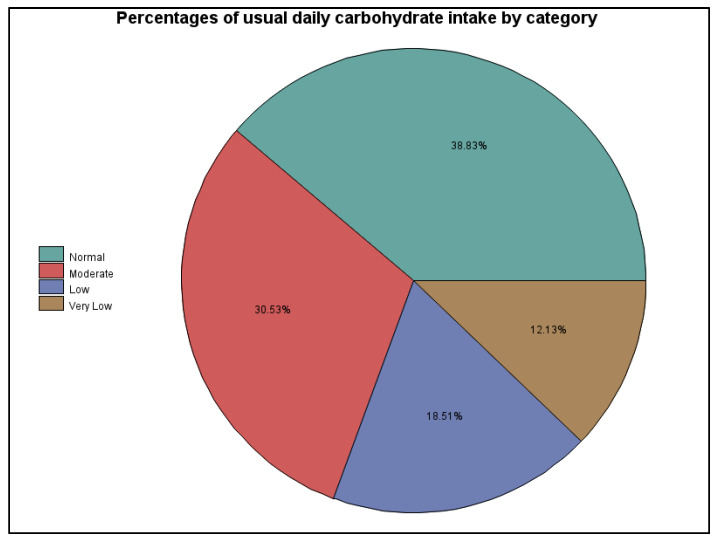
Percentages of usual daily carbohydrate intake by category.

**Table 1 ijerph-18-09332-t001:** Participant Characteristics and Survey Question Responses.

Characteristic or Survey Question	*N*	Mean	Median	SE	Min	Max
Latest A1C (%)	220	6.6	6.6	0.1	4.2	10.5
Age (years)	220	42.1	40	1	13	84
Time with T1D (years)	220	21	18	1	1	80
Total weekly physical activity (minutes)	220	498	360	26	30	2520
Total weekly physical activity (hours)	220	8.3	6	0.4	0.5	42
Days per week of physical activity (number)	220	5.2	5	0.1	2	7
Typical duration of physical activity (minutes)	220	93	75	5	15	720
Carbohydrate intake (1 = normal to 4 = very low)	220	2.1	2	0.1	1	4
Ingest carbs if glucose falls with activity (yes/no)	220	1.07	1	0.02	1 (yes)	2 (no)
Insulin pump use (yes/no)	220	1.4	1	0.03	1 (yes)	2 (no)
Noninsulin diabetes medication use (yes/no)	220	1.86	2	0.02	1 (yes)	2 (no)
Statin use to lower blood cholesterol (yes/no)	220	1.71	2	0.03	1 (yes)	2 (no)
Self-monitor blood glucose (yes/no)	220	1.05	1	0.02	1 (yes)	2 (no)
Continuous glucose monitor use (yes/no)	220	1.22	1	0.03	1 (yes)	2 (no)
Fitness activities (yes/no)	220	1.18	1	0.03	1 (yes)	2 (no)
Endurance sports or training (yes/no)	220	1.29	1	0.05	1 (yes)	2 (no)
Endurance-power sports (yes/no)	220	1.75	2	0.03	1 (yes)	2 (no)
Power sports or training (yes/no)	220	1.86	2	0.02	1 (yes)	2 (no)
Outdoor recreational activities (yes/no)	220	1.53	2	0.03	1 (yes)	2 (no)
Exercise-induced low blood glucose (yes/no)	220	1.13	1	0.02	1 (yes)	2 (no)
Exercise-induced high blood glucose (yes/no)	220	1.32	1	0.03	1 (yes)	2 (no)

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
