# Peer review of "Physical Activity, Dietary Patterns, and Glycemic Management in Active Individuals with Type 1 Diabetes: An Online Survey"

_ijerph, 2021, doi:10.3390/ijerph18179332_

Round 1
Reviewer 1 Report
In this manuscript the authors conducted an online survey on patients with type 1 diabetes about their glycemic management, physical activity patterns, carbohydrate 14 and dietary intake. The study data are self-report and in total anonymity. The idea of the study is interesting but the methodological rigor has many gaps.
Comments:
- the authors studied only self-report data from patients without any consultation with the treating physicians, this represents a great lack of information
- the authors have not established personal criteria that can characterize the variables studied. A 13-year-old patient has different eating habits, physical activity, and complications of diabetes than an 84-year-old patient
- the authors did not consider any other co-morbidities that patients and drugs taken in addition to insulin may have
- the authors studied physical activity by dividing self-reports by patients into static categories. The degree, the intensity, and above all an objective data to compare the various types of sports such as MET/min/week could be taken into consideration. In this way it is not possible to have clear results
- the authors investigated subjects from different geographic areas in the world without considering the different eating habits as well as the different health regimes in the treatment of diabetes
- the authors did not research the effects and complications of diabetes. How many patients have had cardiovascular complications? How many have kidney failure? These are all fundamental data to compare the data
- the manuscript needs a revision of the English language
Reviewer 2 Report
While it can provide interesting information with regards to physical activity and dietary patterns in relation to glycemic management, there are important flaws in the manuscript which should be addressed.
The aim, which is not displayed in the abstract, is not clear enough. I would recommend the authors to split the current aim into specific objectives and to align all the sections of the manuscript (specially statistical analyses and results) according to these objectives. Under my point of view, this would improve the organization and clarity of the manuscript.
Some more specific comments are provided for each of the manuscript sections.
Introduction
Clear as it is, I think the introduction is excessively short and concise. A deeper description of existing evidence is needed to understand the relevance of the present study as well as the hypothesis established by the authors.
Materials and methods
The authors claim that consentment was not requested given the nature of the study. However, informed consent is one of the founding principles of research ethics and it is widely assummed that participants must give consent before they enter the research.
The participants subsection in which the authors characterize the study sample should be included in this section and not in the results one.
In the subsection 2.1.1. the authors explain how they categorized physical activities. However, the do not provide any reference to support these categories. I wonder why basketball or soccer are considered endurance-power sports while volleyball or beach volleyball are included in the power sports group. I have some reservations about the classification given by the authors.
In the subsection 2.2.2. the authors claim that carbohydrate intake was estimated according to the respondents’ answers to open questions. Authors should especify how this estimation was performed (how much carbohydrate was associated with each of the dietary patterns/actual answers provided by the participants?). Again, I have my reservations about the reliability of the carbohydrate intake measure.
In lines 146-150 the authors provide some rationale to use log A1C instead of raw A1C values. However, they display both values in the results section. I think the manuscript would gain in clarity if they authors reported either of these two values.
Results
As stated before, I think the objectives should be reformulated and this section should be reorganized following the aforementioned objectives.
In lines 194-195 the authors claim that participants were categorized in two groups (those reaching recommended PA levels and those not reaching that level). They state in brackets that participants reached recommended levels when performing at least 150 minutes. However, the PA guidelines for americans (cited by the authors) are more precise when establishing the recommended levels (Adults should do at least 150 minutes to 300 minutes a week of moderate-intensity, or 75 minutes to 150 minutes a week of vigorous-intensity aerobic physical activity, or an equivalent combination of moderate- and vigorous-intensity aerobic activity. They should also do muscle-strengthening activities on 2 or more days a week. Older adults should do multicomponent physical activity that includes balance training as well as aerobic and muscle-strengthening activities.). If authors have actually taken these guidelines as a reference, they should be more precise when describing the recommendations in the manuscript. If they have categorized participants solely based on the amount of PA minutes (regardless of the type), I wonder whether we can presume the reliability of the findings.
I do not think the figures clearly display the results and believe that the authors should improve them.
Discussion
This section provides interesting information regarding the findings of the study as well as the evidence from existing literature.
Again, I think that the reorganization of the manuscript following different specific objectives would be helpful to improve the clarity of the work. The discussion section could then also follow the structure established in the objectives and be aligned with them and with the results section.
Lastly, the authors make a good job in the presentation of the study limitations and future lines of study.
Round 2
Reviewer 1 Report
The authors replied to the comments, however many of the gaps in the manuscript remain.
I have no further comments